# Probiotic Enrichment and Reduction of Aflatoxins in a Traditional African Maize-Based Fermented Food

**DOI:** 10.3390/nu11020265

**Published:** 2019-01-25

**Authors:** Alex Paul Wacoo, Ivan Muzira Mukisa, Rehema Meeme, Stellah Byakika, Deborah Wendiro, Wilbert Sybesma, Remco Kort

**Affiliations:** 1Department of Molecular Cell Biology, VU University Amsterdam, De Boelelaan 1108, 1081 HZ Amsterdam, The Netherlands; wacooalex@gmail.com; 2Yoba for Life Foundation, Hunzestraat 133-A, 1079 WB Amsterdam, The Netherlands; wilbert.sybesma@yoba4life.com; 3Department of Nursing, Muni University, P.O. Box 725 Arua, Uganda; 4Department of Food Technology and Nutrition, School of Food Technology Nutrition and Bioengineering, College of Agricultural and Environmental Sciences, Makerere University, P.O. Box 7062 Kampala, Uganda; ivanmukisa@caes.mak.ac.ug (I.M.M.); meemerehema@gmail.com (R.M.); stellahbyakika@gmail.com (S.B.); 5Food and Agriculture Division, Standards Department, Uganda National Bureau of Standards, P.O. Box 6329 Kampala, Uganda; 6Department of Microbiology and Biotechnology, Product Development Directorate, Uganda Industrial Research Institute, P.O. Box 7086 Kampala, Uganda; dwendiro@gmail.com; 7TNO, Microbiology and Systems Biology, Utrechtseweg 48, 3704 HE Zeist, The Netherlands

**Keywords:** fermented cereal beverage, *kwete*, maize, probiotics, *Lactobacillus rhamnosus*, aflatoxins, binding

## Abstract

Fermentation of food products can be used for the delivery of probiotic bacteria and means of food detoxification, provided that probiotics are able to grow, and toxins are reduced in raw materials with minimal effects on consumer acceptability. This study evaluated probiotic enrichment and detoxification of *kwete*, a commonly consumed traditional fermented cereal beverage in Uganda, by the use of starter culture with the probiotic *Lactobacillus rhamnosus* yoba 2012 and *Streptococcus thermophilus* C106. Probiotic *kwete* was produced by fermenting a suspension of ground maize grain at 30 °C for a period of 24 h, leading to a decrease of the pH value to ≤ 4.0 and increase in titratable acidity of at least 0.2% (w/v). Probiotic *kwete* was acceptable to the consumers with a score of ≥6 on a 9-point hedonic scale. The products were stable over a month’s study period with a mean pH of 3.9, titratable acidity of 0.6% (w/v), and *Lactobacillus rhamnosus* counts >10^8^ cfu g^−1^. HPLC analysis of aflatoxins of the water-soluble fraction of *kwete* indicated that fermentation led to an over 1000-fold reduction of aflatoxins B_1_, B_2_, G_1_, and G_2_ spiked in the raw ingredients. In vitro fluorescence spectroscopy confirmed binding of aflatoxin B_1_ to *Lactobacillus rhamnosus* with an efficiency of 83.5%. This study shows that fermentation is a means to enrich with probiotics and reduce widely occurring aflatoxin contamination of maize products that are consumed as staple foods in sub-Saharan Africa.

## 1. Introduction

The sustainable production of traditional foods in sub-Saharan Africa offers a viable opportunity to fight increasing hunger and malnutrition [1]. Cereals such as millet, sorghum, and maize are important sources of food in Africa [2], and are predominantly cultivated for human nutrition, in particular, for children [3,4,5]. The maize-based African fermented *kwete* is a fermented beverage, which is traditionally produced by Luo communities, but now commercially available in many rural and urban areas in Uganda [6,7]. Kwete is a slightly alcoholic, with a thick consistency and a sweet–sour taste [8]. Consumers use these beverages as social drinks, a source of energy, thirst quenchers, and weaning foods [6,8,9]. However, cereals are highly susceptible to aflatoxin contamination. This could be attributed to their rich nutrient composition and relatively high humidity that favors fungal growth [10,11]. Aflatoxin contamination in cereals, such as maize, has been reported to be as high as 46 mg kg^−1^ and 19 mg kg^−1^, in Kenya and Uganda, respectively [11]. These levels of aflatoxin contamination are of great concern considering that maize is used in all pre-primary, primary, and post-primary schools in Uganda for the preparation of breakfast (porridge) and lunch (posho or pap, which is a solid gelatinized product comprising of maize flour and water). The schools get the maize through parental in-kind contributions, direct procurement from the open markets and, to a small extent, from school gardening [12]. None of these sources of maize are subject to quality control and could, therefore, be contaminated with aflatoxins.

There is an urgent need to decrease the risk of aflatoxins due to concerns over human health, food safety, and economic losses. Aflatoxins have created a lot of havoc, particularly in sub-Saharan Africa, where acute and mostly chronic aflatoxicosis has been reported. In 2004, approximately 317 cases of aflatoxicosis and 125 deaths were reported in Kenya [13]. A pilot study in Uganda, which evaluated aflatoxin exposure in rural populations, reported that all 100 adults included in the study and 92 out of 96 children contained detectable levels of aflatoxin–albumin adducts, including five babies who were exclusively fed via breastfeeding [14]. This could explain the high number of cases of liver cancer (estimated at 6.5 and 5.5 age-standardized incidence rate per 100,000 people annually for males and females, respectively) in Uganda [15]. Furthermore, aflatoxins suppress the activity of the human immune system by significantly lowering the levels of perforin, perforin-expressing, and granzyme A-expressing CD8+ T cells. This results in impaired CD^8+^ T cells which, in turn, affects cellular immunity against infectious diseases [16,17]. Aflatoxins also affect absorption of nutrients through alteration of intestinal integrity [18], thus affecting child growth and development. 

Therefore, the mitigation of aflatoxins in food is of great importance, and methods such as high-performance liquid chromatography (HPLC) and enzyme-linked immunosorbent assay (ELISA) have been developed to monitor their levels. However, these methods can only be operated in laboratories by well-trained personnel [19,20]. As a drive to reduce exposure to contaminated food, we developed an on-site detection method [21] and validated this method for the analysis of maize flour from markets and households in Kampala [22]. Using this innovative detection methodology, we found that the average aflatoxin contamination at household level was higher (22.2 ± 4.6 µg kg^−1^) than at the markets (7.6 ± 2.3 µg kg^−1^). About three out of four samples from households tested positive for aflatoxins and nearly half the samples contained higher levels than the East Africa acceptable limit of 10 µg kg^−1^ [23]. This is a strong indication that high levels of aflatoxins are consumed daily by the Ugandan population. A combinatory approach aimed at prevention of contamination, lowering the amount of aflatoxins in the food chain as well as decreased uptake after human ingestion, could be most effective in lowering the burden of aflatoxins in cereals.

Fermentation has been appreciated as means to bring down the concentration of aflatoxins in cereal-based foods [24]. However, variations in product quality and safety associated with the undefined nature of starter cultures in traditional fermentations create a barrier for this approach. The application of defined starters with lactic acid bacteria could offer an easy and inexpensive method that could be adopted for detoxifying aflatoxins in food in a controlled and reproducible manner [25,26]. The potential of probiotic strains to reduce the risk of aflatoxins has been studied, both in vitro and in vivo [27,28]. Probiotics do not only mitigate aflatoxins but also convey other health benefits to consumers, such as the decrease of intensity and duration of diarrhea, which is a major disease burden, especially in developing countries [29]; a promising candidate in this respect is the probiotic model strain *Lactobacillus rhamnosus* GG [30]. 

The probiotic *L. rhamnosus* GG is now accessible in East Africa, under the name *L. rhamnosus* yoba 2012, following the introduction of the concept of ‘generic probiotics’ [31,32]. The *L. rhamnosus* yoba 2012 strain has been previously applied for the preparation of African traditional products like *uji* (fermented maize), *mutandabota* (fermented pulp of the baobab fruit and milk), *zomkom* (fermented wheat), and the naturally fermented milk *lait caillé* [1,31,33]. Recently, Di Stefano et al. [34] reported the growth parameters, organoleptic characteristics, and acceptability of fermented millet by use of *L. rhamnosus* GR-1 and *S. thermophilus* C106, which provided a good reference point for this study. The inclusion of probiotic starters in a product may affect the products’ sensory properties and, hence, acceptability [35,36,37]. It is therefore not only essential to ensure growth of the probiotic, but to also compare sensory characteristics and consumer acceptability of the probiotic-enriched, traditional fermented products.

In this paper, we evaluated the ability of *L. rhamnosus* yoba 2012 and *S. thermophilus* C106 to propagate in *kwete*, a traditional maize-based drink. In addition, we studied survival of the probiotic during storage and compared consumer acceptability with commercial traditional products previously introduced on the market. We also monitored the effect of the probiotic starter culture on the levels of aflatoxins during fermentation, and confirmed the ability of the probiotic to bind aflatoxin B_1_. The results of this study demonstrate the potential of utilizing widely consumed locally available traditional foods as carriers for probiotics, which adds health benefits and improves product safety.

## 2. Materials and Methods

### 2.1. Ingredients for Probiotic Kwete

The dried probiotic starter culture comprising *L. rhamnosus* yoba 2012 and *S. thermophilus* C106 [31] was obtained from the Yoba for Life Foundation (Amsterdam, The Netherlands) and stored at −40 °C prior to use. Hulled maize flour for preparation of probiotic *kwete* was purchased from Maganzu millers, Kampala, Uganda, and kept at room temperature in a dry place prior to use.

### 2.2. Preparation of Probiotic Kwete

Probiotic *kwete* was prepared using a modification of the traditional method described by Muyanja and Namugumya [8]. To prepare one liter of probiotic non-alcoholic *kwete*, 150 g of hulled maize flour was mixed with 100 mL of water to form dough. The dough was roasted to golden brown on a saucepan over a hotplate with continuous turning to prevent burning. The roasted dough was subsequently mixed in one liter of boiled water to form a slurry. The slurry was boiled for 15 minutes to form porridge and to inactivate all unfavorable microorganisms. The porridge was subsequently cooled down to 45 °C followed by inoculation with prefermented maize porridge of 2% (v/v). The porridge was left to ferment at 30 °C for 24 h, with the acidity and pH monitored at t = 0, 3, 6, 9, 12, and 24 h. Bacterial counts were taken at t = 0, 12, and 24 h. The experiment was carried out in triplicate on separate days. The shelf stability of probiotic *kwete* was determined by physicochemical, microbiological, and consumer acceptability tests. Samples were fermented for 24 h and stored in sterile bottles under refrigeration at 4 °C. Analyses were done at weekly intervals for a period of four weeks. 

### 2.3. Inoculation Approaches

In this study, we evaluated the production of probiotic *kwete* by the use of three inoculation approaches for fermentation at 30 °C. These approaches included (i) prefermented milk (ii) direct inoculation with the dried starter culture, and (iii) prefermented maize porridge. For the first procedure with prefermented milk, one gram of probiotic dried starter culture was used for one liter of milk and incubated overnight at 37 °C, as described previously [38]. Subsequently, the *kwete* base was inoculated with 2% (v/v) of the prefermented milk. The second inoculation was performed by direct addition of one gram of the dried starter culture to one liter of *kwete* base. The third inoculation procedure includes 2% prefermented maize porridge. This porridge was obtained by 50 g of hulled maize flour mixed with one liter of water to a final concentration of 5% (w/v). The mixture was boiled to obtain a thick porridge followed by cooling down to 45 °C and inoculation with one gram of dried starter culture. Fermentation was carried out at 30 °C for 24 h. Samples during fermentation after all three inoculation procedures were taken at t = 0 and 24 h for analysis of pH, acidity, and colony forming units of *L. rhamnosus* yoba 2012 and *S. thermophilus* C106. 

### 2.4. Enumeration of Colony Forming Units

Serial dilutions of probiotic *kwete* samples were prepared by using four-times-diluted Ringer’s solution. Counts of lactic acid bacteria (LAB) were determined by pour plating of selected serial dilutions in sterile MRS agar for *L. rhamnosus* yoba 2012 and M17 agar for *S. thermophilus* C106, followed by incubation anaerobically at 30 °C for 48 h. Yeast counts were determined by surface spreading of selected dilutions in potato dextrose agar and incubating at 30 °C for 2–5 days. Coliform counts were determined by pour plating selected serial dilutions in violet red bile lactose agar and incubation at 30 °C for 48 h. All media were obtained from CONDA Laboratories (Madrid, Spain).

### 2.5. Determination of Titratable Acidity and pH 

Titratable acidity was determined by weighing 10 mL of the *kwete* sample. The sample was subsequently filtered through Whatman (Whatman International Ltd, Maidstone, England) number 1 filter paper. The filtrate was titrated against a standardized solution of 0.1 N NaOH with phenolphthalein as the indicator [39]. The experiment was performed in triplicate. The pH was determined using a bench top FiveGo digital pH meter (Mettler Toledo, Greifensee, Switzerland) which was calibrated using DKD-certified buffers (Mettler Toledo) of pH 4.00, 7.00, and 9.20 prior to analysis. The pH of the samples was determined in duplicate.

### 2.6. Consumer Acceptability of Probiotic Kwete

A panel of 62 Ugandan students (equal ratio males and females) evaluated the acceptability of the probiotic *kwete* one day after fermentation in a double-blind study. The acceptability of the probiotic was compared with non-probiotic, traditionally prepared *kwete* [8]. Panelists ranked their acceptability of various attributes using a 9-point hedonic scale [40]. Water bottles were provided to rinse the palate in between tasting of samples. The group means differences between probiotic and traditionally made *kwete* (control) were analyzed using a *t*-test. All statistical analyses were performed using XLSTAT software (version 2012.4.03, Addinsoft, Paris, France).

### 2.7. Aflatoxin Analysis

#### 2.7.1. Total Aflatoxin Concentration in Kwete

To evaluate the potential of *L. rhamnosus* yoba 2012 in mitigating the risk effect of aflatoxins in a maize-based traditional food, 20 mL of *kwete* base was spiked with 1.25 mL of 120 ng mL^−1^ total aflatoxins (40.0 ng mL^−1^ B_1_, 40.0 ng mL^−1^ G_1_, 20.0 ng mL^−1^ B_2_, and 20.0 ng mL^−1^ G_2_). The aflatoxin standards were obtained from Bioo Scientific (Austin, Texas, USA). Control experiments were set as follows: unfermented *kwete* without aflatoxins, fermented *kwete* without aflatoxins, unfermented *kwete* base with aflatoxins, unfermented *kwete* base with 0.92% lactic acid (pH 4.4) with aflatoxins, and *kwete* spiked with aflatoxins after fermentation. Incubations were carried out for 24 h at 30 °C with samples taken at t = 0, 12, and 24 h for aflatoxin B_1_, B_2_, G_1_, and G_2_ quantification. The water-insoluble phase of *kwete* was removed by centrifugation at 3000*g* for 20 minutes at room temperature. The supernatant was applied to an immunoaffinity column according to the instructions of the manufacturer (AFLASTAR^TM^ R Romer Labs Inc, Union, Missouri, USA). Briefly, 0.5 mL of the extract containing aflatoxins (B_1_, B_2_, G_1_, and G_2_) was diluted to 2.5 mL with deionized water prior to clean-up using the immunoaffinity column. The column was then washed with 4 mL of 16% methanol to remove any unbound aflatoxins, and the bound aflatoxins were eluted using 2 mL absolute methanol. Aliquots of 100 µL of the extract were injected into the HPLC column equilibrated with methanol. The aflatoxins were eluted using a methanol/acetonitrile/water (8:27:65 (v/v)) solution at a flow rate of 0.7 mL/min. Detection and quantification were performed by a fluorescence detector operated at excitation and emission wavelengths of 365 nm and 450 nm, respectively.

#### 2.7.2. Aflatoxin B_1_ Binding to *Lactobacillus rhamnosus*

In order to further substantiate the mechanism for the reduction of aflatoxins during the controlled fermentation of *kwete*, the binding affinity of *L. rhamnosus* yoba 2012 to aflatoxin B_1_ was determined. Briefly, the probiotic *L. rhamnosus* yoba 2012 was cultured in de Man, Rogosa, Sharpe (MRS) broth with 0.1% (v/v) Tween 80 at 37 °C in an atmosphere of air containing 5% CO_2_ for 24 h. The cell pellet was collected by centrifugation at 3200*g* for 10 minutes at room temperature, and washed twice with physiological saline to remove excess MRS broth. The cell pellet was resuspended and serially diluted with physiological saline to generate nine different concentrations in optical densities ranging from 0 to 1 using Ultrospec 2100 pro spectrophotometer (Amersham Biosciences, Piscataway, New Jersey, USA) set at 600 nm. These dilutions were prepared in a microtiter plate and centrifuged to remove supernatant prior to aflatoxin B_1_ binding. An aflatoxin B_1_ solution in physiological saline of 1.0 µg mL^−1^ was added to bacterial cell pellets in a microtiter plate, and the cell suspension was incubated at 37 °C for 30 minutes, followed by centrifugation at 3200*g* for 10 minutes at room temperature. The residual aflatoxin B_1_ in the supernatant was analyzed by fluorescence spectroscopy using the Fluostar Omega microplate reader (BMG Labtech, Ortenberg, Germany) operating with a 390 nm excitation filter and a 480 nm emission filter. The fluorescence of the residual aflatoxin B_1_ was plotted versus the cell concentrations. Curve fitting was carried out with GraphPad Prism version 7 (GraphPad Software, San Diego, CA, USA).

## 3. Results

### 3.1. Fermentation of Kwete Using the Yoba Starter Culture

The *L. rhamnosus* yoba 2012 and *S. thermophilus* C106 bacteria propagated well in the kwete base with notable changes in pH and acidity (Figure 1). Bacterial growth resulted in lactic acid production and an increase in titratable acidity from 1.8‰ to 7.0‰, and a decrease of the pH from 6.2 to 3.9 after 24 h of fermentation (Figure 1). Fermentation of the probiotic cereal-based kwete with a pH of 4.4 and an acidity of 4.5‰ after twelve hours is relatively slow compared to milk fermentation with the same bacterial starter culture [38].

### 3.2. Acceptability of Probiotic Kwete

The consumer acceptability scores of probiotic kwete in comparison to traditionally made kwete assessed by a panel of 62 university students are shown in Table 1. The acceptability scores for color, aroma, and overall acceptability of probiotic kwete were generally comparable (*p* > 0.05) to the local (traditional) kwete on the market. Although the taste of probiotic kwete was highly acceptable, it was quite different from the locally made type, which had very sour and slightly alcoholic flavors with a pH of 3.4 ± 0.1. The latter pH value was similar to those reported by Muyanja and Namugumya [8]. The acceptability scores for probiotic kwete ranged from 3 to 9 (‘dislike moderately’ to ‘like extremely’). 

### 3.3. Shelf Stability of Probiotic Kwete

Shelf stability for probiotic *kwete* was evaluated by monitoring changes in pH, acidity, viability of *L. rhamnosus* yoba 2012, *S. thermophilus* C106, coliforms, yeasts, and consumer acceptability during refrigerated storage for a period of four weeks (Figure 2). The pH of the products ranged between 3.2 and 4.0, and the acidity ranged between 0.6% and 0.7% (*p* < 0.05) during four weeks of storage. *Lactobacillus rhamnosus* yoba 2012 cell counts remained above 10^8^ cfu g^−1^ during the entire storage period, but *S. thermophilus* C106 dropped 3 log units after 2 weeks to 10^4^ cfu g^−1^ in week 4. Coliforms, yeasts, and molds were not detected (<1 cfu g^−1^) in the samples. Probiotic *kwete* remained acceptable during the four weeks of storage with overall acceptability scores ranging, on average, between 6.0 to 7.7 (equivalent to ‘like moderately’ to ‘like very much’). Overall acceptability scores did not vary significantly (*p* > 0.05) during storage.

### 3.4. Effect of Inoculation Method 

The preparation of fermented milk with the freeze-dried starter yoba starter culture bacteria has been widely applied throughout Uganda in areas with access to milk. However, fermented milk as starter culture to ferment kwete base would not be helpful for people living in areas with limited milk availability. Therefore, two alternative approaches for initiating the fermentation of probiotic *kwete* were evaluated (Table 2). Both inoculation approaches (direct use of the dried starter culture and fermented maize porridge) have been used to produce probiotic kwete at room temperature for 24 h. The changes during fermentation in pH, titratable acidity, *L. rhamnosus* yoba 2012, and *S. thermophilus* C106 counts are shown in Table 2. The probiotic bacterium *L. rhamnosus* yoba 2012 performed well for all three fermentations with an increase of two to three log units. The bacterium *S. thermophilus* C106 propagated well in *kwete* inoculated with fermented milk, showing an increase of more than two log units. However, in the absence of milk, in the case of the dried starter and prefermented kwete inoculation methods, the counts only increased slightly and remained at seven log units. For all fermentations, the pH dropped from 5.6–6.3 to 4.2–3.9, and the acidity increased from 0.2% to 0.3%–0.5%. The highest pH and acidity differences were observed for the *kwete* fermentation inoculated with prefermented maize porridge. 

### 3.5. Reduction Aflatoxins B_1_, B_2_, G_1_, and G_2_ by Fermentation

The reduction of aflatoxins B_1_, B_2_, G_1_, and G_2_, spiked into the maize *kwete* base, was assessed during fermentation for 24 h with the yoba starter culture by HPLC analysis. The chromatogram in Figure 3 shows the concentration of aflatoxins from the immunoaffinity-purified water-soluble fraction at t = 0, 12, and 24 h. A notable decrease was recorded in the concentration of aflatoxins B_1_, G_1_, B_2_, and G_2_ of 92% ± 0.1%, 91.4% ± 0.2%, 91.8% ± 0.2%, and 90.9% ± 0.2%, respectively, after a period of 12 h of fermentation. However, after 24 h of fermentation, no detectable levels of aflatoxins were left in the sample, showing that yoba starter culture bacteria efficiently removed the concentration of all of the four major types of aflatoxins (Figure 3). In order to exclude that the reduction of aflatoxins was a result of other factors than the fermentation by the starter culture, we carried out a number of control experiments. These experiments indicated that no aflatoxins can be detected in case they are not added to the raw ingredients, that fermentation with the starter culture is required for the reduction, and that an incubation of 24 h in an acidic environment (lactic acid, pH 4.4) does not lead to a reduction of the four major aflatoxins (Table 3).

### 3.6. Binding of Aflatoxin B_1_ by Lactobacillus rhamnosus

The binding of the major aflatoxin B_1_ to the probiotic bacterium *L. rhamnosus* yoba 2012 was studied by monitoring the residual aflatoxin after incubation of a dilution series of cell suspensions with aflatoxin B_1_ at a concentration of 1.0 µg mL^−1^. The fluorescence aflatoxin B_1_ was plotted as a function of OD_600_ (Figure 4). The residuals for the curve fit at low optical density values (0.0–0.3) of *L. rhamnosus* yoba 2012 were relatively high, probably resulting from high experimental errors at relatively low bacterial cell concentrations (Appendix A). The bacterial cell concentration at OD_600_ of 0.5 showed binding of 83% of the aflatoxin.

## 4. Discussion

Introduction of bacterial probiotic strains in traditional fermented foods can be used as a means to convey their health benefits [41]. In this study, we used the probiotic model bacterium *L. rhamnosus* GG, since there is a wealth of scientific evidence showing its beneficial effects in the prevention and treatment of gastrointestinal diseases, including rotavirus and *Clostridium difficile*-associated diarrhea, and travelers’ and antibiotic-associated diarrhea (AAD) [42,43,44,45]. In addition, this strain is readily accessible in its generic form, *L. rhamnosus* yoba 2012, packed in a lyophilized state together with *S. thermophilus* C106 in a sachet as the yoba starter culture [31]. 

The yoba starter culture bacteria successfully fermented the traditional maize-based food *kwete*, as evident from the production of lactic acid shown by a decrease in pH and simultaneous increase in titratable acid. As required for microbiological safety and stability of lactic acid-fermented beverages [46,47,48,49], the observed pH values of probiotic fermented *kwete* were ≤4.3, and the amount of titratable acid was at least 0.6% after 24 h of fermentation at 30 °C. It should be noted that in case of natural *kwete* fermentations, it can take between 24 to 120 h to attain these pH and acidity values, while— in line with our findings—with defined starter cultures, these values are reached within 12 to 24 h of fermentation [6]. The maximum acidity levels observed during storage of probiotic *kwete*, of 0.7%, corresponded to the maximum levels of acid production previously observed with starters containing *L. rhamnosus* GG for fermentation of maize porridge with added barley [50].

In this study, *kwete* was used as a substrate to enhance growth of the probiotic *L. rhamnosus* yoba 2012, reaching a maximum of 1.0 × 10^9^ cfu g^−1^ after 24 h fermentation of *kwete* at a temperature of 30 °C. These counts of colony forming units were similar to those reported for other traditional products serving as a substrate for the same starter culture, including *mutandabota* (a dairy product containing baobab pulp), *uji* (fermented maize and sorghum beverage), and *zomkom* (a fermented sorghum beverage [1,31]. Maximum counts of *L. rhamnosus* yoba 2012 in *kwete* were also comparable to those reported for other starter cultures with lactic acid bacteria, such as *L. reuteri*, *L. acidophilus* (LA5 and 1748), and *L. rhamnosus* GG in maize porridge [50]. The ability of *L. rhamnosus* yoba 2012 to grow in cereal bases, such as *kwete*, is attributed to the availability of sugars such as glucose, fructose, and maltose from maize and millet/sorghum malt for *kwete* [8], in addition to free amino nitrogen from cereal malt [51]. The traditional production of *kwete* with an undefined mixture of yeasts, *Lactobacillus*, and *Lactococcus* species often results in poor product quality and short shelf life, requiring a consumption of *kwete* within 24 h after production [52]. In probiotic *kwete* prepared by the use of the yoba starter culture, we did not identify any (harmful) coliforms, yeasts, and molds in the samples during 4 weeks of storage.

Different probiotic starters uniquely affect the flavor profile, sensorial properties and, ultimately, the acceptability of products in which they are introduced. For instance, mild acidity, relatively high amounts of acetaldehyde, and the presence of the human isolate *L. plantarum* NCIMB 8826 correlated with higher acceptability scores of barley- and oat-based probiotic beverages [53]. In another study, *L. rhamnosus* LRB and *L. acidophilus* PRO produced probiotic *mageu* (a fermented maize beverage), whose sensory properties and acceptability scores were closer to that of the control than the product produced by *L. casei* BGP1 and *L. paracasei* BGP93 [36]. Therefore, it is necessary to establish the effect of the addition of probiotics on the acceptability of traditional fermented foods. Benchmarking the new probiotic product with existing related traditional products helps in gauging the success of the probiotic product [49]. This study shows that use of the yoba starter did not significantly affect the acceptability of *kwete*. The acceptability of sensory characteristics was comparable to the commercial product on the market. The consumers took note of a difference in the taste, but appreciated the probiotic *kwete* for its sweet and sour taste with a mean score of 6.8 ± 1.4 compared to 6.5 ± 1.6 of commercial brand. Therefore, *kwete* produced using the yoba sachet culture can be readily accepted and frequently purchased by consumers, thus increasing accessibility of probiotics in Uganda. 

For a product to be considered probiotic, it should contain a minimum of 10^6^ cfu per mL or gram of the probiotic microorganisms at the time of consumption [54]. Consuming 100–1000 mL per day of such a product provides the recommended daily dose (10^8^–10^9^ cfu), essential for conveying the health benefits of probiotics [54,55]. Probiotic *kwete* contained a minimum of 4.0 × 10^8^ cfu g^−1^ of *L. rhamnosus* yoba 2012 during four weeks of storage at 4 °C. Thus, a minimum daily consumption of 10 mL of probiotic *kwete* per day would be more than sufficient to meet the recommended daily intake of probiotics. With respect to shelf stability, probiotic *kwete* generally remained stable and acceptable during the entire study period of four weeks. Traditional *kwete* is normally produced and consumed within 24 h [56]. However, the use of yoba starter culture made the product stable for four weeks, thus improving its shelf life. 

Detoxification of aflatoxins in food prior to consumption is a novel approach to curb their toxic effects. Several technologies have been employed to eliminate or reduce levels of aflatoxins in food, but only a handful are accepted for use and, as of yet, none offer 100% efficiency [57]. The use of microorganisms to detoxify aflatoxins has been given more consideration [25,27]. In this study, yoba starter culture bacteria, which are being used extensively in Uganda, Kenya, and Tanzania to produce fermented milk, demonstrated an excellent ability to reduce aflatoxins, during fermentation of *kwete* base, to non-detectable levels. There was a notable reduction in total concentrations containing all four major aflatoxins (B_1_, B_2_, G_1_, and G_2_) from 7.0 ng mL^−1^ to non-detectable levels (Figure 3). The detoxification of aflatoxins in *kwete* could be the result of binding as well as of degradation, as binding alone would not reduce the toxin from the food substrate to non-detectable levels [58]. We speculate that aflatoxin degradation is a specific property of our starter culture, as other studies reported less than 100% removal by *L. rhamnosus* strains [59,60,61]. However, under the experimental conditions used so far, we have not been able to confirm degradation of aflatoxins by pure cultures of bacterial strains in the yoba starter culture. 

Our in vitro fluorescence experiments did confirm binding of aflatoxin B_1_ to a cell suspension of *L. rhamnosus* yoba 2012 at OD_600_ of 0.5, which reduced the aflatoxin B_1_ concentration of 1.0 µg/mL to 17%. Preliminary results indicated that the binding of aflatoxin B1 to *S. thermophilus* C106 was less efficient, with a reduction of aflatoxin B_1_ to 86% at the same cell density. Aflatoxin binding to lactic acid bacteria was previously suggested as a safe means to reduce the bioavailability and enhance excretion of the toxin from the body [62,63]. Although the mechanism of binding is still poorly understood, cell surface polysaccharide, peptidoglycans, and teichoic acids have been suggested as the binding sites [59,64,65]. Here, we show that the yoba starter, including *L. rhamnosus* yoba 2012 and *S. thermophilus* C106, were able to remove 100% of 120 µg kg^−1^ total aflatoxins spiked in the water-soluble fraction of *kwete*, which is highly relevant considering the range of aflatoxin concentrations we previously found in maize flour in households in Uganda [22]. 

## 5. Conclusions

This study showed that yoba starter culture bacteria were able to produce *kwete* products with comparable acceptability to commercially available traditional products. The yoba starter culture bacteria were able to ferment *kwete*, reducing the pH to below 4.0 in 24 h at room temperature. The products remained stable during refrigerated storage for a month. This study demonstrated that yoba starter culture bacteria can reduce aflatoxins during fermentation to non-detectable levels. Accordingly, fermentation with this starter culture can positively contribute to reduction of the risk of aflatoxins in maize-based foods widely used in schools in Uganda.

## Figures and Tables

**Figure 1 nutrients-11-00265-f001:**
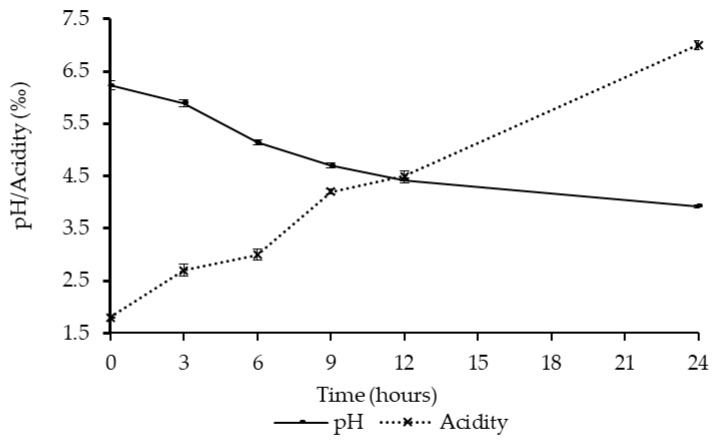
Changes in pH and acidity during fermentation of *kwete* with *L. rhamnosus* yoba 2012 and *S. thermophilus* C106 at 30 °C. Data points represent means of three independent fermentations. Error bars represent standard deviations.

**Figure 2 nutrients-11-00265-f002:**
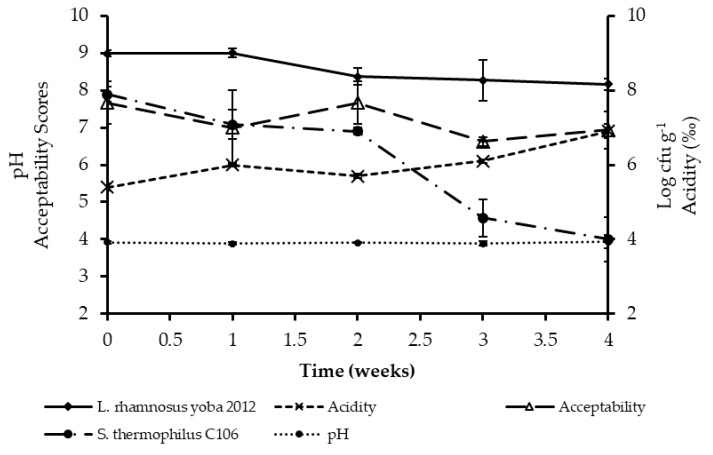
Shelf stability of probiotic kwete containing *L. rhamnosus* yoba 2012 and *S. thermophilus* C106 during storage at 4.0 °C. Data points for pH, acidity, *L. rhamnosus* yoba 2012, and *S. thermophilus* C106 counts are means of three independent fermentations, while data points for acceptability are means of eight panelist scores. Error bars represent standard deviations. A 9-point hedonic scale (1 = dislike extremely and 9 = like extremely) was used for acceptability scoring.

**Figure 3 nutrients-11-00265-f003:**
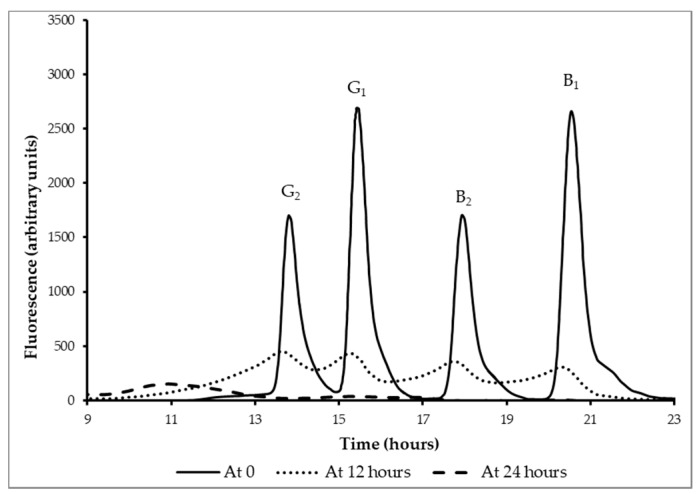
Reduction of aflatoxins B_1_, B_2_, G_1_, and G_2_ during controlled fermentation of the traditional maize-based kwete. HPLC chromatograms (fluorescence at 450 nm (arbitrary units) as a function of time (hours)) of immunoaffinity-purified water-soluble fractions of kwete after 0, 12, and 24 h of fermentation.

**Figure 4 nutrients-11-00265-f004:**
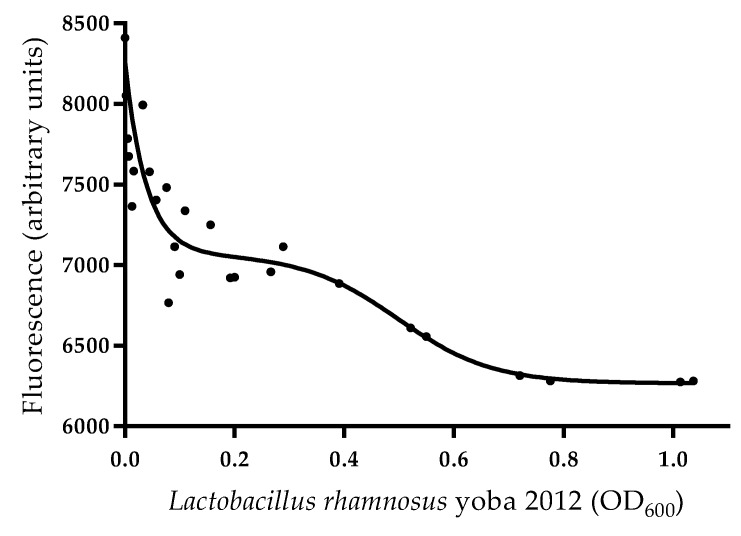
*Lactobacillus rhamnosus* yoba 2012–aflatoxin B_1_ binding curve.

**Table 1 nutrients-11-00265-t001:** Comparison of the consumer acceptability scores of probiotic *kwete* produced by *L. rhamnosus* yoba 2012 and *S. thermophilus* C106 with a commercial brand on a 9-point hedonic scale: 9 = like extremely, 8 = like very much, 7 = like moderately, 6 = like slightly, 5 = neither like nor dislike, 4 = dislike slightly, 3 = dislike moderately, 2 = dislike very much, 1 = dislike extremely. Values are means and standard deviations (*n* = 62 respondents).

	Sample	Acceptability Scores				
		Appearance	Color	Aroma	Taste	Overall
***kwete***	Probiotic	6.2 ± 1.1	6.6 ± 1.9	6.7 ± 0.8	6.8 ± 1.4	6.4 ± 1.7
	Commercial brand	6.8 ± 2.2	6.1 ± 1.2	6.3 ± 2.1	6.5 ± 1.6	6.5 ± 2.0
	*p*-value	>0.05	>0.05	>0.05	<0.05	>0.05

**Table 2 nutrients-11-00265-t002:** Comparison of inoculation methods for initiating the fermentation of *kwete* with *L. rhamnosus* yoba 2012 and *S. thermphilus* C106. Values are means of two independent fermentations (no variations observed in pH values; variation in counts <20%; variation in acidity <10%). Fermentations were carried out at 30 °C

Parameter	Time (hours)	Inoculation Methods for *kwete*		
		Prefermented Milk	Dried Starter	Prefermented Maize
*L. rhamnosus* (log cfu g^−1^)	0	6.5	6.3	5.7
	24	8.4	8.7	8.9
*S. thermophilus* (log cfu g^−1^)	0	6.8	7.1	7.3
	24	9.1	7.8	7.9
pH	0	5.6	6.3	6.2
	24	3.9	4.2	3.9
Acidity (% acid)	0	0.2	0.2	0.2
	24	0.3	0.4	0.5

**Table 3 nutrients-11-00265-t003:** Reduction of spiked aflatoxins B_1_, B_2_, G_1_, and G_2_ after 24 h of incubation of *kwete* base. Values are means and standard deviations of analyses from three independent experiments. * Counts were performed by use of selective de Man, Rogosa, Sharpe (MRS) nutrient agar plates and indicate numbers for *L. rhamnosus* yoba 2012.

	Experimental Runs	Initial pH	Final pH	Initial Counts * (log cfu g^−1^)	Final Counts * (log cfu g^−1^)	Aflatoxin Concentration (ng mL^−1^)				
						B_1_	B_2_	G_1_	G_2_	Total
Controls	no spike, no starter	6.3 ± 0.1	6.3 ± 0.1	0	0	0	0	0	0	0
	no spike, starter	6.3 ± 0.3	4.2 ± 0.1	6.5 ± 0.2	9.4 ± 0.3	0	0	0	0	0
	spike (t = 0 h), no starter	6.3 ± 0.0	6.1± 0.3	0	0	2.4 ± 0.1	1.1 ± 0.0	2.4 ± 0.1	1.1 ± 0.0	7.0
	spike (t = 0 h), lactic acid, no starter	6.3 ± 0.1	4.4 ± 0.2	0	0	2.4 ± 0.1	1.1 ± 0.0	2.4 ± 0.1	1.0 ± 0.5	6.9
	spike (t = 24 h), starter	6.3 ± 0.5	3.9 ± 0.2	6.2 ± 0.4	9.0 ± 0.2	2.4 ± 0.3	0.9 ± 0.1	2.4 ± 0.1	1.1 ± 0.0	6.8
Experiment (spike, starter)	t = 0 h	6.3 ± 0.5	6.1 ± 0.4	6.1 ± 0.5	6.5 ± 0.2	2.4 ± 0.2	1.2 ± 0.1	2.4 ± 0.3	0.9 ± 0.1	6.9
	t = 12 h	6.3 ± 0.1	4.7 ± 0.2	6.1 ± 0.5	7.5 ± 0.2	0.2 ± 0.0	0.1 ± 0.0	0.2 ± 0.1	0.1 ± 0.0	0.6
	t = 24 h	6.3 ± 0.1	4.1± 0.1	6.3 ± 0.6	8.9 ± 0.1	0	0	0	0	0

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
