# Peer review of "Probiotic Enrichment and Reduction of Aflatoxins in a Traditional African Maize-Based Fermented Food"

_nutrients, 2019, doi:10.3390/nu11020265_

Reviewer 1 Report

The present work is dealing with detoxification features of probiotic lactic acid bacteria during fermentation of a traditional maize based African fermented food. The topic is interesting and the work is well designed and performed. My main concerns are related to the exposure. An English language revision is recommended.  

P2 L75-80: it appears unuseful with respect the topic and the results of the manuscript.

P3L114: the role of S. thermophilus C106 is unclear. Throughout the manuscript you always talk about L. rhamnosus.

P3 L139-142: Why does the protocol for kwete production change with respect the incoulum with the fresh starter (P3 L126-128)?

P4 L146: units of presumptive lactic acid bacteria. You have not only cells of L. rhamnosus but also S. thermophilus. How did you distinguish them?

P5 L191: Italic font for L. rhamnosus.

P5 L224-228: confused. Please, rephrase.

P6 L234-240: change Italic font

P7 L264: For the reader the link with yogurt is hard to understand.

Table 3: the value of total aflatoxins in the last line is wrong.

P11 L368: You said that the cell density was above  108 during storage (P6 L251), but here you reported 4*107. Why?

Author Response

Reviewer #1

P2 L75-80: it appears unuseful with respect the topic and the results of the manuscript.

We would like to thank the reviewer for this comment,  but we disagree for two reasons: (i) this section refers to our previous work in this field and puts the current paper in perspective (ii) this section provides important data on the relevance (aflatoxin decontamination in maize products) described in the current paper, as we previously reported on the aflatoxin contamination levels in markets and households in Uganda 

P3L114: the role of S. thermophilus C106 is unclear. Throughout the manuscript you always talk about L. rhamnosus.

We acknowledge this important point and have clarified the role of S. thermophilus C106 throughout the manuscript. Please check P3 L103, 107, P4 L146 (L146 and 147 were removed as this was a repetition), 152, P5 L228, P6 L235, 248, 254, P7 L258, FIGURE 2, L 264, 276, 279, TABLE 2, L413.

P3 L139-142: Why does the protocol for kwete production change with respect the incoulum with the fresh starter (P3 L126-128)?

We clarified the production protocol and inoculation methods for the production of probiotic kwete. See P3 L118, P4 and section 2.3 P4 L133 – L146

P4 L146: units of presumptive lactic acid bacteria. You have not only cells of L. rhamnosus but also S. thermophilus. How did you distinguish them?

This has been corrected appropriately. Please check P4 L146: ‘Counts of lactic acid bacteria (LAB) were determined by pour plating of selected serial dilutions in sterile MRS agar for L. rhamnosus yoba 2012 and M17 agar for S. thermophilus C106 followed by incubation  anaerobically at 30 °C for 48 hours’ ; the cfu counts of L. rhamnosus and S. thermophilus added to the table 2 P8.

P5 L191: Italic font for L. rhamnosus.

L. rhamnosus has been converted to italic font. Check L194

P5 L224-228: confused. Please, rephrase.

P5L228 to 233 has been clarified to ‘The L. rhamnosus yoba 2012 and S. thermophilus C106 bacteria propagated well in the kwete base with notable changes in pH and acidity (Figure 1). Bacterial growth resulted into lactic acid production and an increase in titratable acidity from 1.8 ‰ to 7.0 ‰ and a decrease of the pH from 6.2 to 3.9 after 24 hours of fermentation (Figure 1). It should be noted that the fermentation of the probiotic cereal-based kwete with a pH of 4.4 and an acidity of 4.5 ‰ after twelve hours is relatively slow compared to milk fermentation with the same bacteria [38]. ‘

 P6 L234-240: change Italic font

P6 L239 to 246: the italic font has been removed as proposed

P7 L264: For the reader the link with yogurt is hard to understand.

This section has been rephrased to: ‘The preparation of fermented milk with the freeze-dried starter yoba starter culture bacteria has been widely applied throughout Uganda in areas with access to milk. However, fermented milk as starter culture to ferment kwete base would not be helpful for people living in areas with limited milk availability. Therefore, two alternative approaches for initiating the fermentation of probiotic kwete were evaluated (Table 2).’

Table 3: the value of total aflatoxins in the last line is wrong.

The values of total aflatoxins in the last line in table three have been corrected

P11 L368: You said that the cell density was above  108 during storage (P6 L251), but here you reported 4*107. Why?

P11 L383, the cell density has been corrected to 108 cfu g-1

Reviewer 2 Report

In this manuscript, authors evaluated the probiotic starter culture for fermentation of kwete and for reduction of aflatoxins. The probiotic kwete prepared with the probiotic starter culture showed acceptability and shelf stability. The starter culture removed aflatoxins in the fermentation. In addition, Lactobacillus rhamnosus yoba 2012, a strain of the starter culture, was found to bind aflatoxin B1, suggesting this strain mainly contribute to the removal in the kwete fermentation. This study was conducted well for the evaluation of the starter culture. However, I found several points to be considered for revision.

 A major concern is the counts of L. rhamnosus yoba 2012. The starter culture contains Streptococcus thermophilus C106 as well as L. rhamnosus yoba 2012. However, in most parts of this manuscript, the results were discussed as the effects of only L. rhamnosus yoba 2012. The counting of LAB (L149-150) does not seem to differentiate these strains. How could the effects only by L. rhamnosus yoba 2012 be evaluated? Although the effects of L. rhamnosus yoba 2012 on binding aflatoxin was proved, how was the contribution of S. thermophilus C106 in the fermentation and the binding?

 Specific comments

L147-; Yeasts and coliforms were counted?

L195; MRS medium originally contains 0.1% of Tween 80. Here, additional Tween 80 was used? How was the concentration?

L224 and later; Species names should be italicized.

L234-240; Do not italicize.

Figure 2; Acceptability in the legend of symbols.

Figure 3; Time (hours) should be Time (min)?

Table 3; Initial and final counts were for LAB? Or these counts contain other contaminants such as yeasts and coliforms?

L330; correct reference numbers.

Author Response

A major concern is the counts of L. rhamnosus yoba 2012. The starter culture contains Streptococcus thermophilus C106 as well as L. rhamnosus yoba 2012. However, in most parts of this manuscript, the results were discussed as the effects of only L. rhamnosus yoba 2012. The counting of LAB (L149-150) does not seem to differentiate these strains. How could the effects only by L. rhamnosus yoba 2012 be evaluated? Although the effects of L. rhamnosus yoba 2012 on binding aflatoxin was proved, how was the contribution of S. thermophilus C106 in the fermentation and the binding?

We acknowledge this important point and we have clarified the role of S. thermophilus C106 in fermentation throughout the manuscript. Please check P3 L103, 107, P4 L146 (L146 and 147 were removed as this was a repetition), 152, P5 L228, P6 L235, 248, 254, P7 L258, Figure 2, L 264, 276, 279, TAable 2, L413.

As for the role in aflatoxin binding, we added a remark to the discussion:

‘Preliminary results indicated that the binding of aflatoxin B1 to S. thermophilus C106 was less efficient with a reduction of aflatoxin B1 to 86% at the same cell density (data not shown).’

L147-; Yeasts and coliforms were counted?

Yeast and coliforms were not detected in probiotic kwete. Please check P6 L254 and P7 L258

L195; MRS medium originally contains 0.1% of Tween 80. Here, additional Tween 80 was used? How was the concentration?

MRS medium did not contain 0.1 % of Tween 80, were therefore added 0.1 % (v/v) of Tween 80. (P5 L210)

L224 and later; Species names should be italicized.

L228: The font has been italicized.

L234-240; Do not italicize.

P6L239 to 246: the italic font has been removed as proposed

Figure 2; Acceptability in the legend of symbols.

Figure 2: Acceptability has been corrected.

Figure 3; Time (hours) should be Time (min)?

Unfortunately, Time (hours) cannot be adjusted to Time (min), as these units are the default values for the software linked to the HPLC .

Table 3; Initial and final counts were for LAB? Or these counts contain other contaminants such as yeasts and coliforms?

Counts were performed with use of selective MRS nutrient agar plates and indicate the numbers for L. rhamnosus yoba 2012 (added to the legend of Table 3)

L330; correct reference numbers.

The reference line in L342 has been corrected

Reviewer 3 Report

The key point of this study is that simply treating lactic acid bacteria can reduce the production of aflatoxin. Complex mechanism analysis and sophisticated experimental methods are excluded. It seems that the quality of the research is insufficient to be published in MDPI Nutrients (IF> 4).

Author Response

The key point of this study is that simply treating lactic acid bacteria can reduce the production of aflatoxin. Complex mechanism analysis and sophisticated experimental methods are excluded. It seems that the quality of the research is insufficient to be published in MDPI Nutrients (IF> 4). 

We would like to thank the reviewer for these comments,  but they are difficult to understand for the authors of the manuscript. ‘…. simply treating lactic acid bacteria’, is this referring to the process of fermentation?  Note that reduction of the production of aflatoxins by lactic acid bacteria is not the topic of research in our paper.  Our paper is about decontamination of a maize based food from aflatoxins by controlled fermentation with a dried starter culture. In our opinion this research can be very impactful, as this starter culture became available in Uganda (and other east African countries) over the recent years.  

What is actually the point made by the reviewer ‘Complex mechanism analysis …. are excluded’.  Are these methods lacking in our research paper or should they be removed from our paper?   

(if we understand correctly) We acknowledge the reviewer’s remark about the key point of this study and the reviewer’s concern about complex mechanism analysis and sophisticated experimental methods. We excluded the formulas 1), 2) and 3), complex binding analysis and supplemental file S1 from the manuscript.

Round  2

Reviewer 1 Report

The manuscript has been improved. 

Reviewer 3 Report

The feedback of authors makes sense in informing the scientific importance of their research.